# Evaluation of circulating plasma proteins in breast cancer using Mendelian randomisation

Anders Mälarstig [1,2] ✉, Felix Grassmann[1,3], Leo Dahl [4], Marios Dimitriou [1,2], Dianna McLeod[1], Marike Gabrielson[1], Karl Smith-Byrne[5], Cecilia E. Thomas [4], Tzu-Hsuan Huang [6], Simon K. G. Forsberg [7], Per Eriksson [7], Mikael Ulfstedt[7], Mattias Johansson[8], Aleksandr V. Sokolov[9], Helgi B. Schiöth[9], Per Hall[1,10], Jochen M. Schwenk [4], Kamila Czene[1] & Åsa K. Hedman [1,2]

Biomarkers for early detection of breast cancer may complement population screening approaches to enable earlier and more precise treatment. The blood proteome is an important source for biomarker discovery but so far, few proteins have been identified with breast cancer risk. Here, we measure 2929 unique proteins in plasma from 598 women selected from the Karolinska Mammography Project to explore the association between protein levels, clinical characteristics, and gene variants, and to identify proteins with a causal role in breast cancer. We present 812 cis-acting protein quantitative trait loci for 737 proteins which are used as instruments in Mendelian randomisation analyses of breast cancer risk. Of those, we present five proteins (CD160, DNPH1, LAYN, LRRC37A2 and TLR1) that show a potential causal role in breast cancer risk with confirmatory results in independent cohorts. Our study suggests that these proteins should be further explored as biomarkers and potential drug targets in breast cancer.

Breast cancer is globally the most common cancer in women and is associated with significant morbidity and mortality[1]. Genome-wide and exome-wide genetic association studies have successfully identified over 300 breast cancer susceptibility loci[2–4] but the mechanisms underpinning most loci and specific gene variants remain uncharacterised, which limits the translation of genetic susceptibility loci to new therapies and precision medicine tools[4].

Mendelian randomisation (MR) offers an alternative approach to the mapping and understanding of aetiologically important pathways in cancer risk and development. MR aims to elucidate causal relationships between modifiable risk factors and disease based on the analysis of genetic variants in observational data[5]. In comparison to genome-wide association studies (GWAS), MR exploits a more confined test space, which increases statistical power, and inherently supports causal gene identification. MR can be further supported by genetic colocalization analysis of exposure and outcome[6]. The relevance of MR has been evaluated and supported by retrospective analyses of drug targets with a proven aetiological or causal role in disease from randomised controlled trials (RCT)[7,8].

Circulating proteins possess many of the characteristics suitable for the discovery of breast cancer biology using MR. First, the plasma

[1]Department of Medical Epidemiology and Biostatistics, Karolinska Institutet, Stockholm, Sweden. [2]Pfizer Worldwide Research Development and Medical, Stockholm, Sweden. [3]Institute of Clinical Research and Systems Medicine, Health and Medical University, Potsdam, Germany. [4]Science for Life Laboratory, Department of Protein Science, KTH Royal Institute of Technology, Solna, Sweden. [5]Cancer Epidemiology Unit, Nuffield Department of Population Health, University of Oxford, Oxford, UK. [6]Cancer Immunology Discovery, Pfizer Inc., San Diego, California, USA. [7]Olink Proteomics AB, Uppsala, Sweden. [8]Genomic Epidemiology Branch, International Agency for Research on Cancer (IARC/WHO), Lyon, France. [9]Department of Surgical Sciences, Functional Pharmacology and Neuroscience, Uppsala University, Uppsala, Sweden. [10]Department of Oncology, Södersjukhuset, Stockholm, Sweden. ✉e-mail: anders.malarstig@ki.se

proteome has been shown to reflect both normal physiology and pathogenic biological processes in cancer[9]. Second, circulating proteins can be measured with high throughput and precision using a variety of advanced methods[10,11]. Third, recent studies have shown that a majority of circulating proteins are associated with cis-acting protein quantitative trait loci (pQTL) i.e., located within 1 Mbp from the protein-encoding gene[12,13]. Fourth, individual cis-pQTL explains relatively large proportions of variance in the protein, making them statistically powerful instrumental variables for causal inference using MR[12,14]. Hundreds of pQTL for plasma proteins have been identified, but so far, no studies have reported pQTL in an entirely female population[7,12,13,15–19].

Here, we measured a total of 2929 unique proteins (2949 assays) using the Olink PEA Explore assay in plasma samples taken from 598 women who were free of a breast cancer diagnosis at the time of sampling. We (1) performed a genetic association analysis of protein levels to identify cis-pQTL and (2) used the cis-pQTL as instrumental variables in MR analysis of breast cancer in the Breast Cancer Association Consortium (BCAC) case-control meta-analysis of breast cancer risk, and (3), replicated MR findings in a second breast cancer case-control meta-analysis of FinnGen[20] and the UK Biobank[21]. Lastly, we followed up on significant proteins identified in the MR analysis by visualising and evaluating colocalization of the protein and breast cancer genetic associations and evaluated potential causal relationships with established and emerging breast cancer risk factors, also using MR (Fig. 1).

Out of 730 plasma proteins evaluated using MR, genetically elevated levels of five proteins were associated with breast cancer risk, namely CD160, 2'-deoxynucleoside 5'-phosphate N-hydrolase 1 (DNPH1), layilin (LAYN), Leucine rich repeat containing 37 member A2 (LRRC37A2) and toll-like receptor 1 (TLR1), which were confirmed in an independent set of data. Our results suggest that these five proteins are aetiologically relevant for breast cancer development. Pending further validation, these findings may point to drug target opportunities or stratification biomarkers in breast cancer.

## Results

### Sample characteristics

The KARMA study consented and recruited a total of 70,877 women during mammography screening from two Swedish regions (Stockholm and Skåne). The aim of the project is the identification of risk factors for breast cancer[22]. The sample for the present substudy was selected for the purpose of evaluating plasma protein biomarkers in relation to incident breast cancer within 2 years from blood sampling. The selection included samples from 299 women in the Southern Sweden (Skåne) region who received a breast cancer diagnosis within 2 years after a blood draw and 299 random controls from the same region, who, as of 2021, had remained breast cancer-free. No difference between cases and controls was seen for median age, body mass index or percent women receiving hormone replacement therapy at the time of blood draw. The proportion of smokers and women with a family history of breast cancer was more common among cases (Table 1).

### Protein analysis, detectability, and quality control

We chose to analyse the plasma samples using an affinity proteomics approach. While targeted methods, such as the Olink PEA approach, are inherently biased towards the subset of proteins that are measured, we attempted to maximise the possibility for discovery by measuring as many proteins as possible. Hence, we used the recently launched version of Olink's Explore I and II panels, which includes 2949 proteins (Supplementary Data 1). Out of this set, 2213 (75%) could be detected in >50% of the samples when judging their normalised protein expression levels (NPX) above the limit of detection (LOD) (Supplementary Fig. 1, Supplementary Data 1). The ranges per protein

**Fig. 1 | Flowchart of study design, data analyses and main results.** A total of 2929 unique proteins (2949 assays) were measured using the Olink PEA Explore assay in plasma samples taken from 598 women. Protein levels were correlated with baseline clinical characteristics using linear regression. Genetic association analysis of protein levels was performed which led to the identification of 812 cis-pQTL for 737 proteins, which were used in Mendelian randomisation analysis of breast cancer in the Breast Cancer Association Consortium (BCAC) case-control meta-analysis of breast cancer risk, followed by replication in independent studies of breast cancer risk. To follow up on significant proteins, the genetic signals for protein levels and breast cancer risk were visualised and evaluated using Mirror plots and were also tested for causal relationships with established and emerging breast cancer risk factors, also using Mendelian randomisation.

**Table 1 | Baseline characteristics of women in KARMA who remained free of breast cancer and nested cases who developed breast cancer within 2 years of sampling**

| Variable | Controls (BC negative) | Cases (incident BC) |
|---|---|---|
| Number of individuals | 299 | 299 |
| Age at baseline (S.D) [years] | 58.83 (9.26) | 58.11 (9.49) |
| Body mass index at interview (S.D) [kg/m²] | 25.20 (4.16) | 25.73 (4.14) |
| Hormone replacement therapy ever [%] | 35.66 | 37.76 |
| Current smoker at interview [%] | 11.23 | 16.32 |
| Family history of BC [%] | 11.27 | 20.92 |
| Number of births (S.D) [%] | 2.2 (1.0) | 1.9 (1.0) |
| Alcohol (S.D) [gram/week] | 46 (59) | 49 (54) |

varied between 0.17 NPX and 9.27 NPX (Supplementary Fig. 2). For data analyses, proteins >25% detectability were included. The proportion of proteins above LOD was lower for the most recent addition to the panels (Explore II). However, it is worth noting that the set of proteins in Explore II are, on average, less abundant than those of the Explore I panel, as shown in a comparison of average levels across proteins overlapping with a mass spectrometry peptide-based analysis generated by the Human Protein Atlas effort (Supplementary Data 2, Supplementary Fig. 3)[23].

## Association between plasma protein levels and incident breast cancer

To evaluate the association of proteins with breast cancer risk, a regression model adjusting for age at blood draw, body mass index and sample storage time was fitted for each of the Olink proteins surpassing QC. A 5% false-discovery rate was used to determine statistical significance. None of the proteins surpassed the threshold for statistical significance and therefore, the protein levels from incident cases and controls were analysed jointly (Supplementary Fig. 4).

## Association between plasma protein levels and clinical characteristics

To examine observational relationships between protein levels and clinical characteristics of the KARMA women, we regressed each measured protein against seven factors (age, alcohol consumption, number of births, body mass index (BMI), hormone replacement therapy (HRT), peri- and post-menopause and current smoking). All associations are shown in Supplementary Data 3. A total of 684 proteins were associated with BMI and 459 proteins were associated with age (Fig. 2). Several of the observed associations have previously been described such as higher plasma levels of leptin and fatty-acid binding protein 4 (FABP4) with increasing BMI[24], higher Follicle stimulating hormone (FSHB) in post-menopausal women and higher placental alkaline phosphatase (PLAP) levels in smokers[25]. Some less described correlations included lower plasma levels of glycodelin (PAEP) and chordin like 2 (CHRDL2) and higher levels of glycoprotein hormone alpha polypeptide (CGA) in post- and peri-menopausal women, and lower levels of osteomodulin (OMD) in women using (HRT).

The replication of known trait-to-protein associations suggests that the data quality was satisfactory and that additional trait-to-

protein associations are enabled by the expansion of the number of detectable proteins.

## Identification of cis-pQTL

To identify genetic instruments for the downstream causality testing using MR, gene variants within a range of 1 Mbp up and downstream of genes encoding each of the 2929 unique proteins were tested for association with levels of the corresponding protein. Significant associations (Bonferroni corrected for the number of independent variants tested at $p < 2.77 \times 10^{-4}$, Supplementary Fig. 5) were observed for a total of 812 independent variants ($R^2 < 0.1$) across 737 proteins, henceforth referred to as cis-pQTL (Supplementary Data 4). Most of the pQTL were observed for proteins on Olink Explore I panel ($n = 523$) but several pQTL were also observed for Explore II proteins ($n = 289$). Some of the cis-pQTL showed effect sizes well above 1 standard deviation, including the nucleotidase NT5C (missense, Pro68Leu, MAF 3%), acylphosphatase (ACYP1) (-7 kbp upstream of gene, MAF 1.5%) and carboxypeptidase Q (CPQ) (intron, MAF 1.7%). We conclude that pQTL are readily detected for proteins on both Explore I and II panels, providing potential MR instruments for 737 proteins.

## Replication analysis

To investigate the validity of the cis-pQTL identified in KARMA, effect sizes were compared with cis-pQTL previously reported for a subset of 90 proteins measured using Olink PEA in the SCALLOP CVD-I study[7]. Measurements for all 90 proteins were available in the KARMA study. Of those 90, cis-pQTL for 33 of the proteins reported by the SCALLOP CVD-I study were associated in KARMA at $p < 0.05$. The beta estimates were strongly consistent across all overlapping proteins. The Pearson

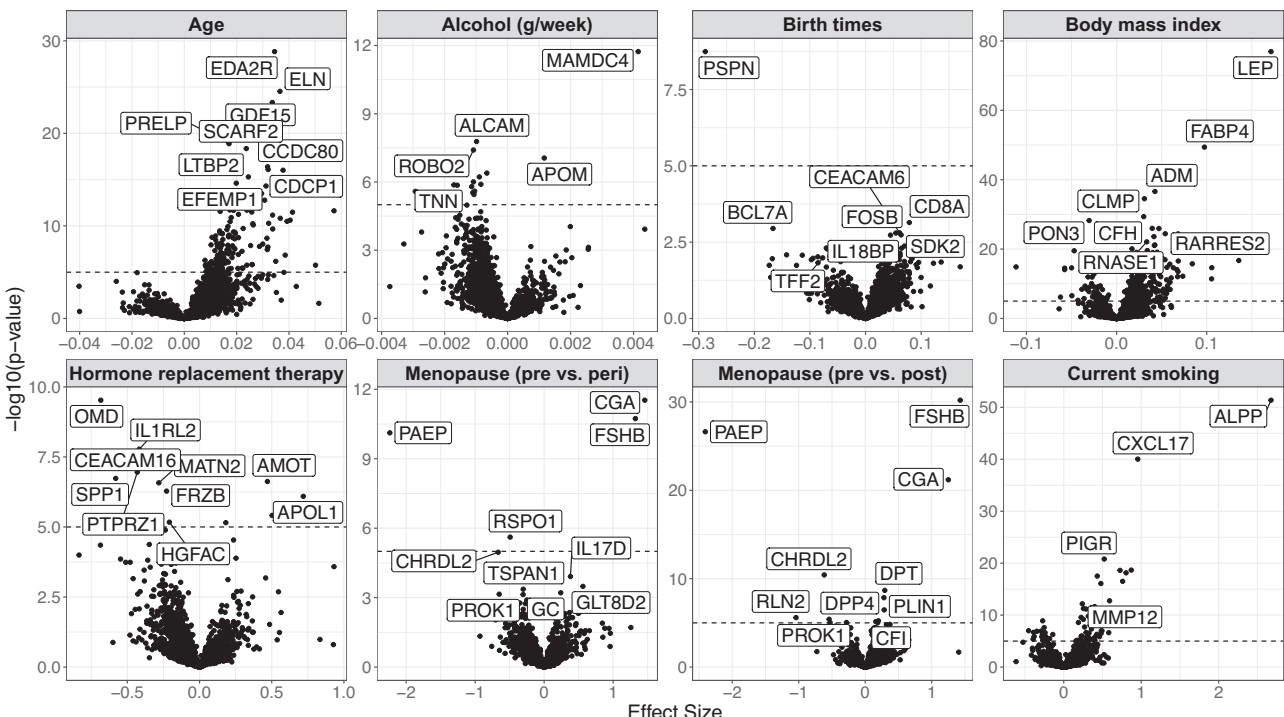

**Fig. 2 | Volcano plots showing estimated effect sizes (x-axis) and the corresponding non-adjusted −log10(p-value) (y-axis) for each of the 2476 proteins analysed in relation to KARMA baseline characteristics.** The plots show estimated effect sizes (x-axis) and the corresponding non-adjusted 2-sided −log10(p-value) (y-axis) with the dashed line marking $p = 1 \times 10^{-5}$ for visual support. Effect sizes are given by a linear regression model per protein, including all 7 baseline characteristics. Each panel shows one of the investigated baseline characteristics,

corresponding to one term in the regression model. The names of up to ten significant proteins per clinical parameter are indicated in each panel according to FDR < 0.05 corrected statistical significance (unadjusted $p < 0.0037$). The number of proteins reaching FDR-adjusted significance were for age: 459, Alcohol consumption: 172, Birth times: 7, BMI: 684, HRT: 93, Menopause pre vs. peri: 18, Menopause pre vs post: 127, Current smoking: 213.

correlation coefficient between effect sizes for the 33 overlapping variants was 0.91 (Supplementary Fig. 6).

To also investigate the generalisability of the identified cis-pQTL, the variants, or those in high linkage disequilibrium (LD) (>0.8), were looked up in previously published studies reporting cis-pQTL based on the Somascan proteomics platform[26,27]. The overlap of Olink proteins available after quality control in the KARMA study and proteins measured in previously published work based on the Somascan platform was 569 proteins (Supplementary Data 4). Of the 603 significant cis-pQTL observed in KARMA for the subset of overlapping proteins, we observed evidence of replication for 374 proteins at Bonferroni-corrected $p < 6.1 \times 10^{-5}$ whereas a total of 229 cis-pQTL did not show evidence of replication at the aforementioned $p$-value threshold.

## Mendelian randomisation analysis

We performed two-sample inverse-variance weighted or Wald-scores MR analysis using protein exposures from the KARMA cis-pQTL to investigate potential causal effects on breast cancer risk using outcome data from BCAC and from the FinnGen R8-UK-biobank meta-analysis[5]. We did not identify genetic proxies for seven of the proteins with cis-pQTL in KARMA, resulting in the testing of 730 protein exposures. Of those, seven proteins surpassed the statistical threshold for significance ($p < 7.5 \times 10^{-5}$) in the discovery study (Fig. 3) of which five replicated in the independent breast cancer case/control study from FinnGen[20] and UK Biobank[21] with consistent effect sizes and directions (Table 2). The replicated proteins, shown here by the names of their encoding genes, were *CD160, DNPH1, LAYN, LRRC37A2,* and *TLR1*. The full summary of MR results is provided in Supplementary Data 5.

We further investigated whether the five proteins with replicated MR evidence for all breast cancers were equally associated with estrogen receptor (ER)-positive compared to ER-negative breast cancer (Table 3). However, the effect sizes were similar across ER+ and ER−

breast cancer risk, suggesting these five proteins are associated equally with ER+ and ER− breast cancer risk.

It was also hypothesised that proteins with MR evidence for an aetiologically important role in breast cancer might influence breast cancer risk via a breast cancer risk factor. To test this, further MR analysis was performed using GWAS of potential breast cancer risk factors as outcomes, including age at menarche, age at menopause, waist-hip ratio, mammographic density, sex hormone binding globulin and insulin growth factor 1 levels (IGF-1)[28]. LRRC37A2 showed MR evidence for later age at menarche and earlier age at menopause in two independent outcome datasets, and also for higher IGF-1 levels (Supplementary Data 2). CD160 showed nominal MR evidence for an aetiological role lower age at menarche.

To summarise, the MR analysis showed that genetic elevation of CD160, DNPH1, LAYN, LRRC37A2 and TLR1 associated with breast cancer risk, and with similar effects on ER+ and ER− cancers.

## Colocalisation analysis

All imputed variants in proximity to the cis-pQTL for proteins with significant MR evidence were visually inspected with the corresponding genomic region for breast cancer risk using mirror plots. The cis-regions around DNPH1 and LRRC37A2 showed the strongest concordance between lead variants for protein levels and breast cancer risk (Supplementary Figs. 8 and 10). Lead pQTL in cis-regions for CD160, LAYN and TLR1 were not the variants with the lowest $p$-values for breast cancer risk but were localised in the same, size-limited, genomic region. We considered the cis-pQTL to be colocalised with breast cancer risk (Supplementary Figs. 7, 9 and 11).

## Systematic search for drugs targeting CD160, DNPH1, LAYN, LRRC37A2 and TLR1

To investigate if any of the five proteins identified in the present investigation had been previously explored as drug targets, we

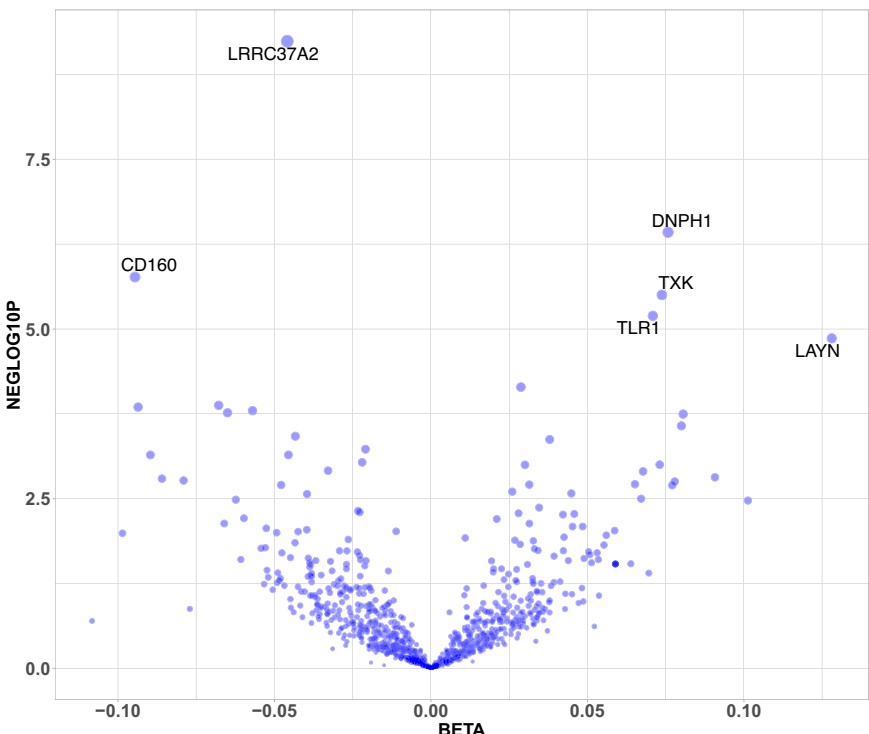

**Fig. 3 | Volcano plot of effect sizes (*X*-axis) and −log10p (*Y*-axis) for the 730 proteins tested for breast cancer risk in the Mendelian randomisation analysis.** Mendelian randomisation analysis on breast cancer risk in the BCAC study was performed by modelling exposure to genetically higher plasma levels of 730 proteins with at least one cis-pQTL. The *Y*-axis shows the −log10 *p*-value of the Wald-score or IVW and the *X*-axis shows the beta-estimates of the MR result for each protein that was tested.

performed a systematic search across several databases, including NIH Pharos Consortium, IUPHAR/BPS Guide to Pharmacology, DrugBank and ClinicalTrials.gov. With the exception of LAYN, targeted by Hyaluronic acid, none of the proteins were registered as known drug targets[29].

## Discussion

We measured 2929 circulating proteins in plasma from 598 women to identify 812 independent cis-pQTL which were applied in MR to investigate associations between genetically predicted protein levels and breast cancer risk. We found that genetically lower levels of CD160 and LRRC37A2 and genetically higher levels of DNPH1, LAYN and TLR1 were associated with increased risk of breast cancer. In addition, genetically higher levels of LRRC37A2 are associated with age at menarche, which adds to previous knowledge of its modest MR evidence for breast cancer risk[28]. MR using cis-pQTL instruments models life-long genetic exposure to higher or lower protein levels, which implies an aetiologically important role of associated proteins in disease. However, similar to some other breast cancer risk factors, plasma levels of the circulating proteins implicated by the MR analysis did not associate with the 2-year risk of breast cancer at the observational level. We cannot exclude that future studies that are larger or that include longitudinal pre-diagnostic samples may uncover such associations.

Among the five proteins identified in our study, DNPH1, also described as Rcl, encodes the enzyme 2'-deoxynucleoside 5'-phosphate N-hydrolase, which plays a role in nucleotide metabolism and is a target of ETV1 -a transcription factor expressed in breast tumours[30]. Two independent CRISPR screens for modulators of BRCA-associated breast tumour sensitivity to PARP inhibitors, an established treatment in BRCA-deficient breast cancer, have shown that genomic inhibition DNPH1 sensitises BRCA-deficient cells to treatment with PARP inhibitors[31,32]. The lead pQTL identified in KARMA, rs75591122, is located ~18.2 kbp upstream from the DNPH1 gene on chromosome 6 and is one of several variants proximal to the DNPH1 gene associated with DNPH1 gene expression levels across multiple tissues[33]. Genetically increased circulating protein levels of DNPH1 were in our study associated with increased breast cancer risk, which is concordant with experimental studies suggesting that DNPH1 inhibition in breast cancer may be a promising avenue for drug development.

Another of the five proteins was CD160, a receptor expressed in immune cells that has been described to play important roles in NK cell biology, predominantly functioning as an activating NK-cell receptor[34]. CD160 is predominantly expressed by healthy NK cells and is one of the driver genes for a specific NK subset related to higher cytokine production[35]. Reduction in CD160 expression led to impaired NK cells and poor outcomes in Hepatocellular carcinoma patients[36]. Since dysfunctional NK cells also correlate with breast cancer progression[37], it can be hypothesised that CD160 could have a similar protective role in breast cancer. Indeed, in our study, genetically elevated circulating protein levels of CD160 are associated with a protective effect in breast cancer, suggesting that a drug activating CD160 specifically on NK cells may enhance anti-tumour immune responses in breast cancer.

Our search for drug targets highlighted the connection between LAYN and Hyaluronic Acid. LAYN encodes Layilin, which is a talin-binding transmembrane and integral membrane protein functioning as a receptor for Hyaluronic acid (HA), with a role in cell adhesion and motility[38,39]. HA is an extracellular matrix component that impacts the tumour microenvironment where elevated HA levels have been reported in multiple cancer types, including breast cancer[40]. Interestingly, targeted depletion of HA controlled the breast cancer tumour growth in xenotransplant mouse models of immunocompetent mice but not of immunodeficient mice, which indicates a potential tumour-immunity role for its receptors, i.e., Layilin[41]. Accordingly, high LAYN expression belongs to transcriptomic signatures specific for regulatory T cells (Tregs) and exhausted CD8+ T cells for several cancer types including breast cancer[42,43]. In our study, the genetic elevation of LAYN protein levels is associated with increased breast cancer risk, suggesting a LAYN inhibitor would be desired for the treatment of breast cancer. However, mechanistic studies will be required to

## Table 2 | Results of the Mendelian randomisation analysis for breast cancer risk

| Exposures | BCAC, all breast cancer | | | FinnGen and UK-Biobank | | |
|---|---|---|---|---|---|---|
| Protein | nsnp | beta | pval | nsnp | beta | pval |
| CD160 | 1 | −0.09 | 1.70E−06 | 1 | −0.07 | 1.50E−02 |
| DNPH1 | 1 | 0.08 | 3.80E−07 | 1 | 0.05 | 3.50E−02 |
| LAYN | 1 | 0.13 | 1.40E−05 | 1 | 0.12 | 8.40E−03 |
| LRRC37A2 | 1 | −0.05 | 5.70E−10 | 1 | −0.05 | 6.80E−05 |
| MST1 | 1 | 0.03 | 7.20E−05 | 1 | 0.02 | 6.60E−02 |
| TLR1 | 1 | 0.07 | 6.40E−06 | 1 | 0.11 | 7.40E−05 |
| TXK | 1 | 0.07 | 3.10E−06 | 1 | 0.03 | 3.40E−01 |

Exposures indicate the protein that was tested. The nsnp indicates the number of instrumental variables (variant associated with protein level) used in the Mendelian randomisation analysis. The beta value is the Mendelian randomisation causal estimate. The causal estimates shown are Wald ratios, which were calculated for each instrumental variable as the beta-value for breast cancer risk divided by the beta-value for the protein level. To estimate two-sided p-values, the Wald ratio standard error was first calculated as the standard error for breast cancer risk divided by the beta-value for the protein level whereupon the z-score was calculated, and the p-value computed from standardised normal distribution at each tail. BCAC stands for data from Breast Cancer Association Consortium[2].

## Table 3 | Mendelian randomisation analysis for estrogen-receptor-positive and negative breast cancer risk

| Exposures | ER+ breast cancer | | | | ER− breast cancer | | | |
|---|---|---|---|---|---|---|---|---|
| | BCAC | | FinnGen | | BCAC | | FinnGen | |
| Protein | beta | pval | beta | pval | beta | pval | beta | pval |
| CD160 | −0.08 | 5.10E−04 | −0.14 | 6.90E−03 | −0.06 | 9.30E−02 | −0.07 | 2.80E−01 |
| DNPH1 | 0.08 | 6.20E−06 | 0.07 | 8.80E−02 | 0.09 | 6.00E−04 | 0.05 | 3.40E−01 |
| LAYN | 0.12 | 5.50E−04 | 0.13 | 1.20E−01 | 0.12 | 2.60E−02 | 0.17 | 1.00E−01 |
| LRRC37A2 | −0.04 | 1.80E−06 | −0.06 | 3.50E−02 | −0.04 | 7.90E−03 | −0.01 | 8.30E−01 |
| TLR1 | 0.07 | 1.60E−04 | 0.11 | 4.10E−02 | 0.09 | 2.30E−03 | 0.11 | 9.40E−02 |

Exposures indicate the protein that was tested. The nsnp indicates the number of instrumental variables (variant associated with protein level) used in the Mendelian randomisation analysis. The beta value is the Mendelian randomisation causal estimate. The causal estimates shown are Wald ratios, which were calculated for each instrumental variable as the beta-value for breast cancer risk divided by the beta-value for the protein level. To estimate two-sided p-values, the Wald ratio standard error was first calculated as the standard error for breast cancer risk divided by the beta-value for the protein level whereupon the z-score was calculated, and the p-value computed from standardised normal distribution at each tail. BCAC stands for data from the Breast Cancer Association Consortium[2].

confirm the direction of effect proposed by the MR evidence and to validate LAYN as a drug target in breast cancer.

Several other studies have investigated the genetic elevation of circulating proteins to identify potential aetiological or causal factors for breast cancer risk. Murphy et al. reported that genetically elevated circulating insulin growth factor levels (IGF-1) were associated with a weak but significantly increased risk of breast cancer, whereas IGF-binding protein-3 was unassociated[44]. Zhu et al. demonstrated an absence of association with breast cancer for genetically elevated levels of C-reactive protein[45] and Shu et al. reported a wider MR analysis, instrumenting 1469 proteins using Somascan-based pQTL in the INTERVAL cohort, of which genetic instruments for 26 proteins were found to be associated[45,46]. Bouras et al. instrumented 47 inflammatory cytokines and reported that genetically increased levels of CXCL1 and decreased levels of MIF associated with breast cancer[47]. Of the 28 proteins previously reported in breast cancer MR studies, our study included post-QC data on 22 proteins and a cis-pQTL was identified in our study for five of them (RELT, ENG, TFPI, ISLR2, SCG3). We replicated cis-pQTL reported for RELT, ENG and SCG3 in ref. 26, which were based on the Somascan protein assay, but were unable to replicate pQTL reported for TFPI, ISLR2 (Supplementary Data 4). None of the five proteins surpassed statistical significance for breast cancer risk in our MR study, although SCG3 and TFPI showed nominal significance at the discovery stage (Supplementary Data 5). The lack of replication for RELT and ENG may be explained by differences in instrumental variable selection and statistical thresholds used.

Our study has both strengths and limitations. One of the strengths is the large number of proteins tested for cis-pQTL and that the cis-pQTL used to instrument genetic elevation using MR were identified in women only, which should provide better estimates in MR for female breast cancer. Another strength is that the protein exposures meeting statistical significance in our discovery MR, using data from the BCAC consortium as outcome, were replicated in the independent case-control analysis that combined breast cancer cases and controls in FinnGen and the UK-Biobank.

However, our study had a limited sample size for discovering cis-pQTL with smaller effect sizes. Therefore, we cannot exclude that additional proteins on the Olink Explore II panels harbour significant cis-pQTL but remained undetected in the KARMA sample. In addition, several of the cis-pQTL with very large effect sizes such as ENTPD6 and NT5C, have a minor allele frequency of less than 3% and their effect sizes may well be inflated because of the so-called "winner's curse. To decrease the false-negative error rate we only included variants in cis to decrease the multiple-test burden and corrected the p-value threshold for significance for the number of independent variants in each cis-region. Effect sizes observed in KARMA were highly concordant with an overlapping set of 33 cis-pQTL for proteins measured with Olink PEA that were previously reported in a study several times larger than the present study. To evaluate the robustness of cis-pQTL identified in KARMA, we sought replication for an overlapping set of 569 proteins measured with Somascan. Of those, 2/3 (374/569) were replicated, which is on par with the expected replication rate given differences in protein analysis methods[16].

In conclusion, by applying an MR approach to a broad range of circulating proteins we found that genetically elevated CD160, DNPH1, LAYN, LRRC37A2 and TLR1 were associated with breast cancer. This suggests that these five proteins play an aetiological or causal role in breast cancer, providing a basis for further functional evaluation of their potential as drug targets.

## Methods
### KARMA study collection
The KARMA cohort consists of 70,877 women performing a screening or clinical mammogram at four hospitals in Sweden during the period October 2010–March 2013. Women consented to both risk and prognosis of breast cancer including collection and storage of questionnaire data, mammograms, matched health care register data, and biological samples. The study was approved by the Stockholm Ethical Review Board, https://etikprovningsmyndigheten.se/en/[22]. From KARMA we identified 299 women diagnosed with breast cancer, which occurred within 2 years of blood draw, and who were residents in Southern Sweden. We used the matchit function from the MatchIt library implemented in R to match 299 controls from the KARMA study to the incident cases by randomly drawing women without incident breast cancer so that the median age at blood draw in cases and controls were similar (median matching). Blood samples were collected at baseline. All blood samples were handled in accordance with a strict 30-h cold-chain protocol, which required that all blood samples were transported on ice and were processed and aliquoted within 30 h from the draw. The sample collection included 16 plasma aliquots, one aliquot of extracted DNA and two aliquots of whole blood for backup. In total, EDTA blood samples from 69,440 (98% of the total cohort) study participants were collected.

### Plasma protein measurements on Olink Explore
Plasma proteomics was performed in samples from the 299 BC cases and 299 BC free controls from KARMA, using the Olink Explore I and II panels (Olink Proteomics AB, Uppsala, Sweden) according to the manufacturer's protocol. Explore combines the Proximity Extension Assay (PEA) technology with Next-generation sequencing (NGS).

In brief, the PEA technology uses matching pairs of oligonucleotide-labelled antibody probes. The PEA probes bind to target antigens producing a binding complex where the complimentary oligonucleotides exist in close proximity to each other, enabling the formation of a target sequence. The dual targeting of probes has been proven to produce outstanding specificity enabling a high degree of multiplexing while maintaining sensitivity and a broad dynamic range. In the Olink Explore protocol, the target sequence is amplified in a double PCR reaction and purified before the NGS. The sequence data is processed and normalised to produce Olinks relative quantification unit Normalised Protein eXpression (NPX). The produced DNA signal functionally works as a proxy for the protein levels present in the sample. Further details on the Olink Explore protocol and internal quality control are available in the Supplementary Methods 1 document.

### Olink analysis quality control
The Olink QC-system includes negative controls, used to monitor the background noise and to set the limit of detection (LOD). Supplementary Fig. 1 and Supplementary Data 1 show the percentage of samples with NPX above LOD.

A principal component analysis of all data was performed to detect outliers and to inform potential sample exclusions. Of the 598 samples that were included in the analysis, one sample was excluded entirely, two samples were excluded from the analysis of Explore ONC-II and two samples were excluded from the analysis of the Explore INF-II panel data (Supplementary Methods 1).

### Association with clinical characteristics
For each of the 2949 measured protein levels, the following linear regression model was fitted: NPX ~ age + bmi + menopause_preVSperi + menopause_preVSpost + birth_times + hrt_status + alcohol_gram_week + smoking_status where menopause_preVSperi contrasts pre- versus peri-menopausal patients, menopause_pre VS post contrasts pre- versus post-menopausal patients, hrt_status contrasts current users of hormone replacement therapy versus patients who have never used it or who have used it in the past, and smoking status contrasts current smokers versus those who have never smoked or smoked in the past. All p-values were FDR corrected for the $2949 \times 7$ performed tests.

## Protein QTL mapping

Genome-wide genotyping in the KARMA study was performed using the Illumina iSelect or Oncoarray arrays, followed by imputation using the Wellcome Trust Sanger Institute imputation service using the 1000 genomes phase 3 as reference. Standard quality control was applied as previously described. Variants with a minor allele frequency <0.01 were filtered out prior to analysis. The final dataset included 9087 million variants.

Proteins >75% of NPX values below LOD were filtered out before the pQTL analysis, yielding a total of 2476 proteins in the analysis. Values below LOD were included. The pQTL discovery analysis was performed using an additive model with adjustments for age, BMI and 10 genetic PCs in PLINK 2.0. To preserve statistical power for pQTL identification, only variants within a 1 mega-base pair window of the protein-coding gene were tested for association with respective circulating protein levels. To manage multiple test correction, while limiting false negatives, the total number of variants per cis-region were calculated as well as the number of independent variants ($R^2 < 0.1$). The average number of variants per cis-region was 6249 (Supplementary Fig. 6) and 180 independent variants (min,max 12-511). Statistical significance was therefore defined as an alpha of 0.05 divided by 180 to account for the average number of independent variants tested per cis-region ($p = 2.77E-04$). A false-discovery rate (FDR) at 5% provided a similar estimate ($p < 5.54E-04$).

## Mendelian randomisation analysis

We performed Two-sample MR using the R package TwoSampleMR[48,49] (https://mrcieu.github.io/TwoSampleMR/) to test for proteins with a potential causal role in breast cancer. Independent cis-pQTL ($r^2 < 0.001$) were used as instrumental variables (IV), and GWAS of breast cancer risk from the BCAC consortium were used as outcome, which included data from 122,977 breast cancer cases and 105,974 controls. In the case of a single independent IV Wald Ratio was applied, otherwise, inverse-variance weighted estimates were reported. The threshold for statistical significance was defined as ($7.5 \times 10^{-5}$) to account for multiple testing. The replication analysis was performed in a meta-analysis of FinnGen R9 and the UK-biobank, which included 25,807 cases and 355,307 controls. Only the seven proteins that met statistical significance in the BCAC discovery analysis were included in the replication analysis, and hence a nominal p-value of 0.05 was considered statistically significant.

## Reporting summary

Further information on research design is available in the Nature Portfolio Reporting Summary linked to this article.

## Data availability

The cis-pQTL GWAS summary-level data generated in this study have been deposited in the Zenodo data repository under accession code https://doi.org/10.5281/zenodo.8387905, with the URL https://zenodo.org/record/8387905. The individual-level phenotypes, genotypes, and biospecimen from the KARMA study are available under restricted access, as personal data are protected by the European Union General Data Protection Regulation legislation and the Swedish Ethical Review Authority, but can be requested according to the process described at URL https://karmastudy.org/contact/data-access/. For previously published GWAS studies on breast cancer risk and risk factors, the summary-level data are available at https://bcac.ccge.medschl.cam.ac.uk/bcacdata/ or at the MRC IEU OpenGWAS database [https://gwas.mrcieu.ac.uk/]. Access to FinnGen data is available at https://www.finngen.fi/en/access_results whereas access to UK-biobank data can be accessed at https://www.ukbiobank.ac.uk/.

## Code availability

We used publicly available software for most of the analyses in the manuscript (as described in the "Methods" section). Access to other scripts and pipelines is provided through GitHub. Protein QTL mapping was performed in PLINK2.0 (https://www.cog-genomics.org/plink/2.0/). Statistical analyses and figures, listed below, were conducted in R version 4.1.0. Mendelian randomisation analyses were conducted using the R package TwoSampleMR version 0.5.6 (https://mrcieu.github.io/TwoSampleMR/), regional plots were constructed using R package RACER version 1.0.0 (https://github.com/oliviasabik/RACER). Figures were constructed using standard R or ggplot2 version 3.4.0 (https://ggplot2.tidyverse.org/index.html). The scripts and pipelines for quality control, association analyses and Mendelian randomisation are available at https://github.com/Schwenk-Lab/KARMA_pQTL_MR and https://github.com/Olink-Proteomics/publications/tree/main/KARMA_MR_NC.

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

## Acknowledgements

We thank all the participants in the Karma study and the study personnel for their devoted work during data collection. We also want to acknowledge the participants and investigators of the FinnGen study. The data handling and analysis were enabled by resources provided by the Swedish National Infrastructure for Computing (SNIC), partially funded by the Swedish Research Council through grant agreement no. 2018-05973. We also acknowledge funding from the Swedish Research Council (Grant 2022-00584), the Swedish Cancer Society (Grants 22 2207, 19 0267 and 20 0990) (P.H., K.C.), the Stockholm County Council (Grant 20200102) and the Karolinska Institutet's Research Foundation (Grant 2018-02146) (P.H., K.C.) as well as from the Stockholm County Council (FoUI-954555, FoUI-978540) (P.H., J.S.), Olink Proteomics AB (A.M.) and Pfizer Inc (P.H.). Where authors are identified as personnel of the International Agency for Research on Cancer/World Health Organization, the authors alone are responsible for the views expressed in this article and they do not necessarily represent the decisions, policies or views of the International Agency for Research on Cancer/World Health Organization.

## Author contributions

Conceptualisation: A.M., Å.K.H.; Data curation: Å.K.H., F.G., L.D., S.K.F., P.E.; Formal analysis: Å.K.H., S.K.F., P.E.; Funding acquisition: A.M., P.H., J.M.S., K.C.; Sample and study management: P.H., K.C., M.G.; Laboratory: M.G., M.U., C.E.T., L.D., J.M.S.; Writing—original draft: A.M., Å.K.H.; Writing—review and editing: A.M., F.G., L.D., M.D., D.M., M.G., K.S.M., T.H.H., S.K.F., P.E., M.U., M.J., A.V.S., H.B.S., P.H., J.M.S., K.C., Å.K.H.

## Funding

## Competing interests

A.M., Å.K.H., M.D. and T.H. are employees of Pfizer Research and Development. S.K.F., P.E. and M.U. are employees of Olink Proteomics AB. The remaining authors declare no competing interests.
