## [Peer Review File · Nature Communications]

REVIEWER COMMENTS

Reviewer #1 (Remarks to the Author): expertise in computational proteomics

Simply put, the manuscript just isn't terribly compelling. Understandably the goal was laudable and I gather the primary study for blood biomarkers proved negative but for a GWAS study of this kind it is just quite under-powered and it is not at all clear why these markers would be good drug targets. Given that these are SNPs their effect sizes must be modest on risk.

Reviewer #2 (Remarks to the Author): expertise in breast cancer genomics pQTL

Mälarstig et al present a clearly written description of an analysis in which they measure circulating proteins in plasma from women participating in the KARMA study. The method that they use to measure the proteins (Olink PEA Explore) allows them to measure 2,929 pre-defined proteins that have previously been linked to inflammation, cardiometabolic pathways, neurology or oncology. They then look for associations between protein levels and clinical characteristics or genetic variants. They report five proteins as potentially causal role in breast cancer.

Comments

The authors state that the women were selected for evaluating plasma protein biomarkers in relation to incident breast cancer and refer to a companion paper in which this analysis is described. I think it is a shame that they decided to split the data into two separate papers as I think it would have helped to put the results of this paper into context if the results of the other paper had been available.

I would also have liked some details as to how the blood samples for this study were collected, processed and stored and whether this could impact on the results. For example, for the 716 proteins that they couldn't measure (results line 113) was there any association with the time between blood draw and sample processing?

Similarly, was it always the same samples in which the normalised protein expression levels were below the LOD? If so, was there any difference in the proportion of case samples and control samples that consistently showed normalised protein expression levels below the LOD?

In their analysis of plasma protein levels and clinical characteristics, the authors include both cases and controls as they state that there were no significant differences in protein levels found in the companion paper. This is another instance where not having access to the results in the companion paper makes it difficult to interpret these results. I would expect that a protein, say leptin, that is associated with BMI would be associated with breast cancer risk as BMI is known risk factor for breast cancer. However, there is no difference in BMI between the cases and controls included in this analysis (Table 1). In fact, the BMI in the two groups are so extraordinarily similar I wonder whether the cases and controls been matched on BMI?

Clearly the cases and controls have not been matched on smoking status though. If they have been matched, this should be described and does this matching on some variables (BMI and age, possibly) but not others (smoking) complicate the interpretation of the results?

On a similar point, why are some variables shown in Table 1 (age, BMI, HRT) but not others (alcohol consumption, number of births, menopausal status)?

In their identification of cis-pQTL, the authors cite a threshold for significance of $<2.2 \times 10^{-4}$ (line 148). In the methods, the threshold they cite is $<2.77 \times 10^{-4}$ (line 369). This is based on a calculation that there are, on average, 180 independent variants within 1Mb of each protein coding gene that they analyzed (i.e. $0.05/180$). First, which threshold did they use? Second, shouldn't their threshold take account of the fact that they are testing 2,213 hypotheses (ie 2,123 protein measurements) giving a threshold of $0.05/(180 \times 2213) = 1.25 \times 10^{-7}$? Finally they state that they analysed 2,213 proteins for which the NPX was above the LOD in 50% of their samples (line 113). However, in the methods they say that proteins for which $>75\%$ of samples had NPX that were below the LOD were filtered out, leaving 2,476 for analysis. Again, which threshold for filtering did they use?

Results, line 150. They divide their significant cis-pQTLs into Explore I and Explore II categories. What is the relevance of these categories? Wouldn't it be more informative to divide the results into inflammation, cardiometabolic, neurology and oncology as you would expect that their positive hits would be enriched for oncology, I think.

They identify seven (out of 737) proteins using MR (Table 2). One of these is TXK. However, I can't find TXK in Supplementary Table 1, is this an error?

Colocalization analysis: I found the plots difficult to understand. First there is a trivial error in the numbering. The text says DNPH1 and LRRC37A2 are supplementary Figs 7 and 8 but they are 7 and 9. More importantly, for people who are unfamiliar with these plots, I think the authors need to provide some more explanation either in the methods or the legends. I assumed that the top hits from the protein analysis would always be in red (as in Sup Fig 6 top panel) and that these same variants would

be shown (still in red) but in a different location in the breast cancer analysis (as in Sup Fig 6). In Sup Fig 7, however, the top four dots on the upper panel are orange. Why is this? And why does Sup Fig 9 (LRR37A2) show these horizontal lines for the P values? Is this to do with the LD structure in the region? Or is there something unusual about the distribution of the protein levels?

Discussion: I'm not an expert on MR, but I thought the point was that if you can show a risk between a genetic variant and a measurable intermediate phenotype (eg plasma DNPH1 levels) and between DNPH1 levels and disease risk then you can use the association between the variant and disease to determine whether the association between DNPH1 levels and disease is causal. My concern is that if there is no association between say DNPH1 and breast cancer risk doesn't that undermine the MR concept? More specifically, how can we interpret the statement "Genetically increased circulating protein levels of DNPH1 was in our study associated with increased breast cancer risk" (Discussion line 245) in the light of "This indicates that genetically predicted protein levels did not capture this short-term risk." (Discussion line 234)

Some trivial points:

Discussion, line 224 "We measured 2,949 circulating proteins in plasma". Shouldn't this be 2,929?

T

he order of the Supplementary Tables isn't very helpful. The first one to be cited is Supplementary Table 5 (lines 113 and 115), then Supplementary Table 6 (line 131), then Supplementary Table 1 (line 150) etc.

Reviewer #3 (Remarks to the Author): expertise in Mendelian randomisation

Målarstig et al. present a robust and comprehensive proteomic study of blood plasma from breast cancer patients, utilizing a large sample size (598 women) from the Karolinska Mammography Project. The depth of the study is commendable, analyzing 2,929 unique proteins and identifying potential causal proteins in breast cancer through an ingenious application of Mendelian randomization (MR).

One of the strengths of this study lies in the extensive examination of correlations between protein levels, clinical characteristics, and gene variants. The identification of 812 cis-acting protein quantitative trait loci (pQTL) underscores the rigorous analytical approach employed by the authors and significantly adds to the existing body of knowledge in this field.

The paper effectively uses these pQTLs as instrumental variables in MR analysis, leading to the discovery of five proteins (CD160, DNPH1, LAYN, LRRC37A2, and TLR1) that may play a causal role in breast cancer risk. The authors make a diligent effort to confirm the findings in two independent cohorts (FinnGen R9 and the UK Biobank), strengthening the reliability of the results and providing valuable cross-validation.

Overall, this study represents a significant advancement in our understanding of the blood proteome's relationship to breast cancer. The potential causal proteins identified could be pivotal in the development of future therapeutic strategies.

Although the manuscript is well written, I do have a few comments to be addressed:

1. The paper mentioned several times the “companion paper by Grassmann et al.”, but where is the paper? It is not on the reference list.
2. Although I do believe in the MR-discovered protein targets, I suggest the authors at least try a reverse causality analysis since, for instance, there are more smokers in the cases than in the controls, so the readers (who might not be an expert in human genetics) may question the causal inference.
3. How was the significance threshold for cis-pQTL discoveries determined? It looks like a Bonferroni correction for 812 independent variants (?) If so, were only the 812 variants tested? I would guess all the variants within the 1Mbp window of each coding gene were tested. In this case, the threshold needs to be justified, as the real number of independent tests must be more than 812. Also, in line 149, “independent variants” must correspond to R-squared less than 0.1, not greater than 0.1?
4. In lines 152-154, it is striking to see that some cis-pQTL effects can be “well above 1 SD”. But it looks like these associated variants all have very low MAF. This is thus prone to a winner’s curse. The authors should justify via replications or at least discuss potentially inflated estimates.
5. In the replication analysis, 33 out of 90 CVD-I proteins were reproduced in KARMA. 374 out of 603 proteins were replicated compared to the Somascaan platform. I would like to see further justification for these replication rates, i.e., assuming e.g., the same effect sizes, what kind of proportions out of these proteins shall we expect in the replication samples, given the replication sample size? This will provide us a clue whether the replication rate is reasonable.

6. In the discussion, the 2-year risk results from follow-up data are inconsistent with the MR analysis result (line 231), and the authors attributed it to that genetically predicted protein levels did not capture this short-term risk. Are there any other explanations?

7. Please discuss and explain the reason why none of the previously reported proteins in breast cancer MR studies surpassed statistical significance in this study (line 282).

8. I can't see any legends for Tables 1-3.

9. Please polish Figure 2, e.g. BMI in the figure should be capitalized. It is better to make the figure legend more detailed. It is better to add significance thresholds.

10. The legend of Supplementary Figure 5 is wrongly written as "Supplementary Figure 4A and 4B".

Xia Shen, PhD

Professor in Statistical Genetics

Fudan University

Reviewer #4 (Remarks to the Author): expertise in proteomics

As the latest generation of proteomics platform, the Olink's Proximity Extension Assay (PEA) not only has the high throughput of genomics, but also retains the specificity of protein/antibody recognition. At the same time, it overcomes the limitation of low abundance detection, and overcomes the key factors such as the limitation of detection speed, flux, multiple detection ability and sensitivity of traditional proteomics methods. Especially in the detection of humoral proteome, the use of PEA technology coupled with NGS readout makes it easy to measure the concentration of thousands of human plasma proteins using only a few μL of blood.

The application of Olink's PEA helps to realize the detection of super-sensitive multiple protein markers, unbiased targeted proteomics and precision proteomics, in order to help the discovery of protein markers, drug development, translational medicine, and make "multi-omics integration" truly feasible.

This paper is another successful application of Olink's PEA. Combined with Mendelian randomisation (MR) analysis, authors found Five proteins with a potential aetiological or causal role in breast cancer, providing a basis for further functional evaluation of their potential as drug targets.

**REVIEWER COMMENTS**

**Reviewer #1 (Remarks to the Author): expertise in computational proteomics**

Simply put, the manuscript just isn't terribly compelling. Understandably the goal was laudable and I gather
the primary study for blood biomarkers proved negative but for a GWAS study of this kind it is just quite
under-powered and it is not at all clear why these markers would be good drug targets. Given that these are
SNPs their effect sizes must be modest on risk.

**Response:** We thank the reviewer for providing this perspective and agree that the sample size of the present
study would have been small for a genome-wide association study of a binary disease trait. However, disease
traits are often more complex than circulating protein levels, with the latter showing considerably larger effect
sizes and superior statistical power. Indeed, several previous studies have shown that pQTL can be readily
identified even in moderate samples (PMIDS: 27532455, 25147954). We also agree that common gene
variants, SNPs, for disease traits typically have very small effect sizes, and breast cancer is no exception.

Even with these limitations, our work identified over 800 genetic variants with a robust role in regulating
plasma proteins, which increases the understanding of biological pathways in humans.

In addition, several studies have shown that pQTL mapping and Mendelian randomization can uncover causal
biology that can be exploited to select drug targets (reviewed here:
<https://pubmed.ncbi.nlm.nih.gov/32860016/>)

MR has so far been extensively used to find drug targets in cardiovascular and metabolic disease but less so in
cancer. We hope that the data provided in this manuscript will be used to expand the use of MR in cancer, also
into other female cancers such as ovarian and endometrial cancer. The KARMA data provided here should be a
good basis for such studies, and the summary statistics from the study are shared with the community as part
of this manuscript.

**Reviewer #2 (Remarks to the Author): expertise in breast cancer genomics pQTL**

Mälärstig et al present a clearly written description of an analysis in which they measure circulating proteins in
plasma from women participating in the KARMA study. The method that they use to measure the proteins
(Olink PEA Explore) allows them to measure 2,929 pre-defined proteins that have previously been linked to
inflammation, cardiometabolic pathways, neurology or oncology. They then look for associations between
protein levels and clinical characteristics or genetic variants. They report five proteins as potentially causal role
in breast cancer.

Comments. The authors state that the women were selected for evaluating plasma protein biomarkers in
relation to incident breast cancer and refer to a companion paper in which this analysis is described. I think it is
a shame that they decided to split the data into two separate papers as I think it would have helped to put the
results of this paper into context if the results of the other paper had been available.

**Response:** We thank the reviewer for the kind words and the valuable comment. We confirm that this KARMA
sub-study is indeed a nested case/control design, in which samples were selected from women who suffered
incident breast cancer within 2 years, with controls of similar age and geography. The Olink proteomic data of
that sub-study were analysed with the aim to address two separate questions, which included a) do any of the
2,929 proteins play an aetiological role in breast cancer, and b) do any of the circulating proteins provide a
meaningful contribution to early breast cancer detection.

For question b) we found that after multiple test corrections, none of the proteins were significantly
associated with the 2-year breast cancer incidence. To make this clearer, we have included a paragraph in
"Results" (lines 113-117) to highlight the lack of robust associations with 2-year breast cancer risk but without
going into detail, which we believe would detract the focus of the present manuscript. We have also edited the
previous section on protein associations with breast cancer in the "Discussion" to emphasize the difference
between life-time risk factors and risk factors associated with short-term risk. We also wanted to highlight that
our study was not powered to detect subtle differences, and relied on a sample taken no more than 2 years
before diagnosis. This text was added "However, similar to some other breast cancer risk factors, plasma levels
of the circulating proteins implicated by the MR analysis did not associate with the 2-year risk of breast cancer

at the observational level. We can not exclude that future studies that are larger or that include longitudinal
 pre-diagnostic samples may uncover such associations.”

To still allow reviewers and editors to evaluate this lack of association between proteins per se and 2-year
 breast cancer risk, we also include two tables showing 1) top results from the case/control analysis and 2)
 case/control differences for the five proteins with evidence of causality in the Mendelian randomization
 analysis. None of the proteins reached statistical significance. We decided not to use a Bonferroni-corrected p-
 value threshold as it would likely have been too conservative, with a p-value threshold of 2×10^{-5} . Instead, we
 used an optimised FDR approach to calculate q-values from the GLM analysis. With the FDR approach, a q-
 value of less than 0.05 (corresponding to 5% FDR) was considered significant.

**Table 1. Top 10 proteins associated with 2-year risk of breast cancer using the nested case/control design**
 **and a general linear model with adjustment for age and body mass index. None of the proteins reached 5 %**
 **FDR corrected significance, as shown by the Q-value.**

PROTEIN	PANEL	PERCENT < LOD	ODDS RATIO (95 % CI)			P-VALUE	Q-VALUE
NME3	Inflammation	0%	0.28	0.44	0.69	2.16E-04	0.25
TACSTD2	Oncology	0%	0.32	0.49	0.73	3.22E-04	0.25
NIT1	Cardiometabolic_II	0%	1.39	2.13	3.37	3.47E-04	0.25
PTPRM	Inflammation	0%	0.34	0.50	0.74	4.10E-04	0.25
FSHB	Cardiometabolic_II	9%	0.19	0.36	0.66	6.67E-04	0.28
SERPINA12	Cardiometabolic	0%	1.37	2.21	3.79	6.90E-04	0.28
FGFBP1	Oncology	0%	0.35	0.53	0.79	1.82E-03	0.40
PGLYRP2	Inflammation_II	0%	0.31	0.51	0.79	1.93E-03	0.40
ROBO2	Neurology	0%	0.34	0.53	0.80	2.03E-03	0.40
CDON	Inflammation	0%	0.36	0.55	0.81	2.61E-03	0.41

**Table 2. Five proteins with evidence of causality for breast cancer in the Mendelian randomization analysis**
 **using the nested case/control design and a general linear model with adjustment for age and body mass**
 **index. None of the proteins reached 5 % FDR corrected significance.**

PROTEIN	PANEL	PERCENT < LOD	ODDS RATIO (95 % CI)			P-VALUE	Q-VALUE
CD160	Inflammation	0	0.89	0.62	1.29	0.53	0.71
DNP1	Inflammation	0	1.34	0.90	2.07	0.15	0.60
LAYN	Neurology	0	0.77	0.52	1.15	0.20	0.61
LRRC37A2	Oncology_II	0	0.85	0.59	1.22	0.37	0.67
MST1	Inflammation_II	0.0067	0.53	0.22	1.03	0.07	0.58
TLR1	Inflammation_II	0.005	0.80	0.54	1.16	0.24	0.63
TXK	Oncology_II	0.7426	0.87	0.65	1.24	0.41	0.68

I would also have liked some details as to how the blood samples for this study were collected, processed and
 stored and whether this could impact on the results. For example, for the 716 proteins that they couldn't
 measure (results line 113) was there any association with the time between blood draw and sample
 processing?

Response: We thank the reviewer for this important question. To clarify, we now include in the Methods
 section that “Blood samples were collected at baseline. All blood samples were handled in accordance with a
 strict 30-h cold-chain protocol, which required that all blood samples were transported on ice and were
 processed and aliquoted within 30 hours from draw. The sample collection included 16 plasma aliquots, one
 aliquot of extracted DNA and two aliquots of whole blood for back-up. In total, EDTA blood samples from 69
 440 (98% of the total cohort) study participants were collected.

However, the exact time between blood draw and handling within the 30-hour cold chain window was not
collected, so we are not able to evaluate the relationship between time-from-blood-draw-to-processing and
proportion above LOD. It is worth noting that previous work has shown that protein levels can, besides the
expected degradation, also *increase* with a prolonged time between draw and freezing, rather than *decrease*.
For example, this work by Shen et al. <https://pubmed.ncbi.nlm.nih.gov/29040064/>.

Similarly, was it always the same samples in which the normalised protein expression levels were below the
LOD? If so, was there any difference in the proportion of case samples and control samples that consistently
showed normalised protein expression levels below the LOD?

**Response:** There were no differences in the proportion of samples below LOD and cases and controls, even
before quality control. However, data from three samples presented as technical outliers. For the Reviewers
and Editor, the below figure provides additional granularity on QC. One sample was removed entirely
(2000002989), whereas data points were removed for some combinations of proteins and samples. This
procedure is part of the standard quality control and downstream filtering, before analysis of data. It is now
described in detail in the Methods section as well as in Supplementary Methods 1 “A principal component
analysis of all data was performed to detect outliers and to inform potential sample exclusions. Of the 598
samples that were included in the analysis, one sample was excluded entirely, two samples were excluded
from analysis of Explore ONC-II and two samples were excluded from analysis of the Explore INF-II panel data
(Supplementary methods 1).”

In their analysis of plasma protein levels and clinical characteristics, the authors include both cases and
controls as they state that there were no significant differences in protein levels found in the companion
paper. This is another instance where not having access to the results in the companion paper makes it
difficult to interpret these results. I would expect that a protein, say leptin, that is associated with BMI would
be associated with breast cancer risk as BMI is known risk factor for breast cancer. However, there is no
difference in BMI between the cases and controls included in this analysis (Table 1). In fact, the BMI in the two
groups are so extraordinarily similar I wonder whether the cases and controls been matched on BMI?

Response: We thank the reviewer for this comment. We agree that the rationale for combining incident cases
and controls in the analysis of clinical characteristics can be explained more clearly. We have now added a
section to the Methods section (lines 286-291) of the manuscript stating:

*“From KARMA we identified 299 women diagnosed with breast cancer which occurred within two years of*
*blood draw and who were resident in Southern Sweden and collected at the same hospital. We used the*
*matchit function from the MatchIt library implemented in R to match 299 controls from KARMA study to the*
*incident cases by randomly drawing women without incident breast cancer so that the median age at blood*
*draw in cases and controls was similar (median matching)”*

The controls were thus not matched for BMI, although the average BMI ended up being very similar between
the groups. While the protein leptin was, as expected, strongly associated with BMI, and BMI in turn with
breast cancer risk, we did not find an association of leptin levels with incident breast cancer ($p=0.2$).

Clearly the cases and controls have not been matched on smoking status though. If they have been matched,
this should be described and does this matching on some variables (BMI and age, possibly) but not others
(smoking) complicate the interpretation of the results?

Response: We thank the reviewer for pointing out this detail. The cases and controls were indeed “only”
matched by age, as described in the previous response and as indicate in Table 1 (lines 389-390). The % of
current smokers are clearly different in BC cases and controls. This could in theory confound protein
associations with breast cancer. However, Mendelian Randomization (MR) analysis is robust to such
confounders as we are using genetics to model life-long exposure of different protein levels.

On a similar point, why are some variables shown in Table 1 (age, BMI, HRT) but not others (alcohol
consumption, number of births, menopausal status)?

Response: We thank the reviewer for spotting this error. The missing parameters have now been added to
Table 1 (lines 389-390) “Number of births (S.D) [%]” and “Alcohol (S.D) [gram/week]”.

In their identification of cis-pQTL, the authors cite a threshold for significance of $<2.2 \times 10^{-4}$ (line 148). In the
methods, the threshold they cite is $<2.77 \times 10^{-4}$ (line 369). This is based on a calculation that there are, on
average, 180 independent variants within 1Mb of each protein coding gene that they analyzed (i.e. $0.05/180$).
First, which threshold did they use? Second, shouldn't their threshold take account of the fact that they are
testing 2,213 hypotheses (ie 2,123 protein measurements) giving a threshold of $0.05/(180 \times 2213)=1.25 \times 10^{-7}$?
Finally they state that they analysed 2,213 proteins for which the NPX was above the LOD in 50% of their
samples (line 113). However, in the methods they say that proteins for which $>75\%$ of samples had NPX that
were below the LOD were filtered out, leaving 2,476 for analysis. Again, which threshold for filtering did they
use?

Response: We thank the reviewer for pointing out this mistake. The threshold was indeed set to account for
180 tests per region based on the average number of independent SNPs in each of the cis-regions, defined as 1
Mbp. The threshold provided in the Methods section was thus correct whereas the one in brackets in the
Results section was wrong. This has now been corrected (see results section lines 139-140).

Regarding adjustment for the number of proteins tested, we think this would be overly conservative and lead
to a high false-negative rate; especially considering that there is a strong prior for proteins being regulated in
cis. Previous pQTL studies based on Olink PEA have shown that 80-90 % of the proteins have at least a cis-pQTL
signal at genome-wide significance (e.g Folkersen et al Nat Metab 2020). In addition, a recent preprint based

on Olink Explore I in the UK-biobank, state that “82% of proteins tested (1,162 of 1,425 proteins encoded by
genes on autosomes) had a cis association (within 1Mb from the gene encoding the protein)”, “Genetic
regulation of the human plasma proteome in 54,306 UK Biobank participants” Sun et al, BioRxiv 2022,
<https://www.biorxiv.org/content/10.1101/2022.06.17.496443v1.full>.

We agree that the previous description was clear when describing the 50 % vs. 75 % above LOD filtering. For
QC, an easily interpretable measure is the number of proteins that can be detected in >50 % of samples, which
is indeed stated in the results section. Based on prior experience, we only excluded proteins with >75 % non-
detectable samples. This is now clarified line 107, and we now state that “*For data analyses, proteins >25 %*
*detectability were included*”.

Results, line 150. They divide their significant cis-pQTLs into Explore I and Explore II categories. What is the
relevance of these categories? Wouldn't it be more informative to divide the results into inflammation,
cardiometabolic, neurology and oncology as you would expect that their positive hits would be enriched for
oncology, I think.

Response: Thanks for a great comment. While not described in the results section, supplementary table 2
shows the cis-pQTL breakdown both by Explore I and II categories as well as the individual 384-plex panels e.g.
inflammation, neurology etc. The decision to call out the Explore I vs. II difference was primarily motivated by
the fact that the Explore II proteins have not been widely explore for pQTLs, in contrast to Explore I for which a
preprint based on UK-biobank exists (<https://www.biorxiv.org/content/10.1101/2022.06.17.496443v1.full>).

While the panels were designed to be enriched for proteins with relevance in e.g. oncology, we often find
these categories to be somewhat less relevant for the biology they may reflect. According to Olink Proteomics,
panels are built on proteins that can be measured using a common sample dilution, meaning that disease-
specificity of any given protein is not the primary consideration (www.olink.com). Since the targeted
proteomics approach is not unbiased (proteome-wide), performing pathway enrichment analysis on the
background of the included proteins may be limited in providing insightful details. Many proteins in the assays
either share to be secreted or have been shown to be pleiotropic.

They identify seven (out of **737**) proteins using MR (Table 2). One of these is TXK. However, I can't find TXK in
Supplementary Table 1, is this an error?

Response: This is well spotted. Indeed, TXK had a detectability level of 74.3 % and had erroneously been
filtered out in an earlier iteration of the supplementary table. The correct version of supplementary table 1
(renumbered to ST4) is now provided.

Colocalization analysis: I found the plots difficult to understand. First there is a trivial error in the numbering.
The text says DNPH1 and LRR37A2 are supplementary Figs 7 and 8 but they are 7 and 9. More importantly,
for people who are unfamiliar with these plots, I think the authors need to provide some more explanation
either in the methods or the legends. I assumed that the top hits from the protein analysis would always be in
red (as in Sup Fig 6 top panel) and that these same variants would be shown (still in red) but in a different
location in the breast cancer analysis (as in Sup Fig 6). In Sup Fig 7, however, the top four dots on the upper
panel are orange. Why is this? And why does Sup Fig 9 (LRR37A2) show these horizontal lines for the P
values? Is this to do with the LD structure in the region? Or is there something unusual about the distribution
of the protein levels?

Response: We thank the reviewer for the helpful comments. The numbering has now been corrected (line
192). We agree that the figure needs more explanation and have added the following to the figure legends:
“*The LD information shown is based on LD calculations for the lead pQTL identified in the KARMA cohort, with*
*individual variants shown in supplementary table 1, with the same variants highlighted for breast cancer risk in*
*the BCAC data.*”

The reviewer's interpretation of the Mirror plots is also correct. For DNPH1, the top variant, rs75591122 has a
p-value of 1.13×10^{-10} but was shown in orange. The reason is that another significant variant was chosen for
the LD reference, and was a mistake made at the stage of plotting the data.

Discussion: I'm not an expert on MR, but I thought the point was that if you can show a risk between a genetic
variant and a measurable intermediate phenotype (eg plasma DNPH1 levels) and between DNPH1 levels and

disease risk then you can use the association between the variant and disease to determine whether the
association between DNPH1 levels and disease is causal. My concern is that if there is no association between
say DNPH1 and breast cancer risk doesn't that undermine the MR concept? More specifically, how can we
interpret the statement "Genetically increased circulating protein levels of DNPH1 was in our study associated
with increased breast cancer risk" (Discussion line 245) in the light of "This indicates that genetically predicted
protein levels did not capture this short-term risk." (Discussion line 234)

*Response: We agree with the reviewer that association between a risk factor and disease is a good starting*
*point for MR, mainly because it allows targeted testing of a factor for which you have a prior belief there is a*
*correlative or causal relationship. And this limits the test space. However, MR does not assume, or depend on,*
*known observational relationships between the factor and the disease. In fact, we tested observational*
*relationships between the 2-year risk breast cancer incidence for each of the proteins, and no associations*
*were observed. We now include this information, please see below (and lines 114-117). However, exposure to*
*higher levels of e.g. the contribution of genetics on DNPH1 levels may be too small to be detected in our*
*~300+300 cases and controls, or the time between sampling and breast cancer risk may have been too short to*
*detect the observational relationship. MR is a robust way to model exposure to higher protein levels over the*
*course of life, and we provide near-independent replication data.*

*We have now added a section under results (lines 114-117) stating that "To evaluate the association of*
*proteins with breast cancer risk, a regression model adjusting for age at blood draw, body mass index and*
*sample storage time was fitted for each of the 2,929 proteins. A false-discovery rate <0.05 was used to*
*determine statistical significance. None of the proteins surpassed the threshold for statistical significance (data*
*not shown)."*

Some trivial points:

Discussion, line 224 "We measured 2,949 circulating proteins in plasma". Shouldn't this be 2,929?

*Response: We have updated the number to state 2,929 proteins (line 203). The number 2,949 is for all the*
*assays performed. This includes 20 proteins that were analyzed on two different panels.*

The order of the Supplementary Tables isn't very helpful. The first one to be cited is Supplementary Table 5
(lines 113 and 115), then Supplementary Table 6 (line 131), then Supplementary Table 1 (line 150) etc.

*Response: We have updated the number of the supplementary tables according to the main text.*

**Reviewer #3 (Remarks to the Author): expertise in Mendelian randomisation**

Mälarstig et al. present a robust and comprehensive proteomic study of blood plasma from breast cancer
patients, utilizing a large sample size (598 women) from the Karolinska Mammography Project. The depth of
the study is commendable, analyzing 2,929 unique proteins and identifying potential causal proteins in breast
cancer through an ingenious application of Mendelian randomization (MR).

One of the strengths of this study lies in the extensive examination of correlations between protein levels,
clinical characteristics, and gene variants. The identification of 812 cis-acting protein quantitative trait loci
(pQTL) underscores the rigorous analytical approach employed by the authors and significantly adds to the
existing body of knowledge in this field.

The paper effectively uses these pQTLs as instrumental variables in MR analysis, leading to the discovery of
five proteins (CD160, DNPH1, LAYN, LRR37A2, and TLR1) that may play a causal role in breast cancer risk. The
authors make a diligent effort to confirm the findings in two independent cohorts (FinnGen R9 and the UK
Biobank), strengthening the reliability of the results and providing valuable cross-validation.

Overall, this study represents a significant advancement in our understanding of the blood proteome's
relationship to breast cancer. The potential causal proteins identified could be pivotal in the development of
future therapeutic strategies.

*Response: We thank the reviewer for this great summary and kind words on the work.*

Although the manuscript is well written, I do have a few comments to be addressed:

1. The paper mentioned several times the "companion paper by Grassmann et al.", but where is the paper? It
is not on the reference list.

Response: The Mälärstig et al. manuscript was initially submitted to Nature Communications at the same time
 as our Grassmann et al manuscript. The latter reports on the lack of association between Olink proteins and
 incident breast cancer in the KARMA cohort. The Grassmann et al. manuscript under review in different
 journal, hence we have removed the references. To add some clarity around the observational proteomics on
 breast cancer, we have now added a section in the “Results” (lines 113-117) stating the lack of significant
 findings for 2-year risk of breast cancer.

2. Although I do believe in the MR-discovered protein targets, I suggest the authors at least try a reverse
 causality analysis since, for instance, there are more smokers in the cases than in the controls, so the readers
 (who might not be an expert in human genetics) may question the causal inference.

Response: We agree with the reviewer, and considering the small sample size, we decided to focus this
 manuscript on discovery cis-pQTL, defined as within 1 Mb from the protein coding gene. While we agree that it
 may yield some further insights, we currently do not have the data to run bi-directional MRs based on
 smoking.

However, and for the Reviewers and Editor, we investigated the effect of smoking per se on all proteins,
 including the 5 that we found causal evidence for in breast cancer. None of the 5 proteins showed an
 association with smoking status.

ASSAY	ESTIMATE	STD.ERROR	STATISTIC	P.VALUE	P.VALUE_ADJUSTED
CD160	-0.0009	0.0718	-0.0129	0.9896	0.9992
DNP1	0.1431	0.0759	1.8859	0.0598	0.3314
LAYN	0.0661	0.0472	1.3999	0.1621	0.5548
LRRC37A2	-0.0191	0.0752	-0.2536	0.7998	0.9549
TLR1	-0.0268	0.0473	-0.5660	0.5716	0.8739

3. How was the significance threshold for cis-pQTL discoveries determined? It looks like a Bonferroni
 correction for 812 independent variants (?) If so, were only the 812 variants tested? I would guess all the
 variants within the 1Mbp window of each coding gene were tested. In this case, the threshold needs to be
 justified, as the real number of independent tests must be more than 812. Also, in line 149, “independent
 variants” must correspond to R-squared less than 0.1, not greater than 0.1?

Response: We thank the reviewer for this valid observation. Given the focus of the discovery on cis-pQTL, we
 pulled out all variants within the 1 MBp regions of all genes coding for the proteins measured on the Olink 3k.
 We then calculated the average number of independent variants ($R^2 < 0.1$) across the regions. This number
 was ~180. The per locus p-value threshold was then $0.05/180$ i.e. 2.77×10^{-4} , which was then used for the
 discovery analysis (line 139 in revised manuscript). The number 812 refers to the total number of significant
 independent associations across the 737 proteins showing signal.

Thank you for pointing out the error on line 149. It has now been corrected, see line 140 in the revised
 manuscript.

4. In lines 152-154, it is striking to see that some cis-pQTL effects can be “well above 1 SD”. But it looks like
 these associated variants all have very low MAF. This is thus prone to a winner’s curse. The authors should
 justify via replications or at least discuss potentially inflated estimates.

Response: We agree with the reviewer and have added to the discussion (lines 269-271 in the revised
 manuscript): “In addition, several of the cis-pQTL with very large effect sizes such as *ENTPD6* and *NT5C* have a
 minor allele frequency of less than 3 % and their effect sizes may well be inflated because of the so called
 “winner’s curse.”

5. In the replication analysis, 33 out of 90 CVD-I proteins were reproduced in KARMA. 374 out of 603 proteins
 were replicated compared to the Somascan platform. I would like to see further justification for these
 replication rates, i.e., assuming e.g., the same effect sizes, what kind of proportions out of these proteins shall
 we expect in the replication samples, given the replication sample size? This will provide us a clue whether the
 replication rate is reasonable.

Response: We thank the reviewer for these suggestions. The sample sizes for the replication datasets were
several times larger than the KARMA study, and included both women and men. Considering the difference in
samples size, it is reasonable to assume that any pQTL detected in KARMA should also be detectable in the
replication studies, unless there is a strong interaction effect with gender. Rather than focusing on p-values for
the comparison with the Olink pQTL from SCALLOP CVD-I, we wanted to show the consistency of effect sizes
across KARMA and the SCALLOP data. We have now added (lines 152-153 in revised manuscript) that “*The*
*beta estimates were strongly consistent across all overlapping proteins.*” In addition to stating that 33 out of 90
proteins had a $p < 0.05$. The consistency of the effect size is probably more interesting here, considering the
large difference in sample size ~ 600 vs. $\sim 22,000$. We also added to the Discussion a line to point out that the
pQTL data we compared with were derived “in a study several times larger than the present one” (Lines
272/273)

Regarding the Somascan comparison, there are significant differences between the Olink and the Somascan
assay with regards to detection principles (antibodies vs. aptamers). Nevertheless, we thought it would be
interesting for the reader to see how the novel KARMA cis-pQTL compare with the Somascan-based pQTL.
Again, the smallest study used for the Somascan replication was several times larger (nearly 30,000 for
Ferkingstad and 10,000 for Pietzner et al). The details on overlapping assays and proteins/pQTL that replicated
are shown in columns 25 (with column name: “*Replicating*”) and 26 (with column name:
“*SomascanV4_Assay_available*”) in supplementary table 4 (in the current version). The latter column will tell
whether there was indeed an overlapping assay, which becomes the denominator for checking the replication
rate.

6. In the discussion, the 2-year risk results from follow-up data are inconsistent with the MR analysis result
(line 231), and the authors attributed it to that genetically predicted protein levels did not capture this short-
term risk. Are there any other explanations?

Response: We thank for reviewer for bringing up this question. We believe that the main reason the
observational analysis did not recapitulate the MR findings is the relatively subtle genetic influence on protein
levels. This is mainly meaningful over the course of life, such as what we model with MR. With sensitive
methods and samples collected at multiple time points prior to the breast cancer diagnoses, such changes may
be observed also at the proteomic level but the nested case/control design we used here was likely
insufficient. The statistical power to detect observational changes may also have been limited, given a sample
size of 299 incident BC cases and the same number of controls.

7. Please discuss and explain the reason why none of the previously reported proteins in breast cancer MR
studies surpassed statistical significance in this study (line 282).

Response: We thank the reviewer for this suggestion and have now elaborated on the replication data of
previous MR findings in the Discussion. The edited text is as follows: “Of the 28 proteins previously reported in
breast cancer MR studies, our study included post-QC data on 22 proteins and a cis-pQTL was identified in our
study for five of them (RELT, ENG, TFPI, ISLR2, SCG3). We replicated cis-pQTL reported for RELT, ENG and SCG3
in reference 26, which were based on the Somascan protein assay, but were unable to replicate pQTL reported
for TFPI, ISLR2 (Supplementary table 4). None of the five proteins surpassed statistical significance for breast
cancer risk in our MR study, although SCG3 and TFPI showed nominal significance at the discovery stage
(Supplementary table 5). Lack of replication for RELT and ENG may be explained by differences in instrumental
variable selection and statistical thresholds used.”

8. I can't see any legends for Tables 1-3.

Response: We have now added titles for Tables 1-3, please see updated manuscript version (lines 389-394).

9. Please polish Figure 2, e.g. BMI in the figure should be capitalized. It is better to make the figure legend
more detailed. It is better to add significance thresholds.

Response: We agree with the reviewer and have updated figure 2, including lines for significance to help the
reader and fixed the headlines for reach plot.

10. The legend of Supplementary Figure 5 is wrongly written as “Supplementary Figure 4A and 4B”.

Response: Thank you for spotting this error. We have corrected the legend.

**Reviewer #4 (Remarks to the Author): expertise in proteomics**

As the latest generation of proteomics platform, the Olink's Proximity Extension Assay (PEA) not only has the
high throughput of genomics, but also retains the specificity of protein/antibody recognition. At the same
time, it overcomes the limitation of low abundance detection, and overcomes the key factors such as the
limitation of detection speed, flux, multiple detection ability and sensitivity of traditional proteomics methods.
Especially in the detection of humoral proteome, the use of PEA technology coupled with NGS readout makes
it easy to measure the concentration of thousands of human plasma proteins using only a few μL of blood.

The application of Olink's PEA helps to realize the detection of super-sensitive multiple protein markers,
unbiased targeted proteomics and precision proteomics, in order to help the discovery of protein markers,
drug development, translational medicine, and make "multi-omics integration" truly feasible.

This paper is another successful application of Olink's PEA. Combined with Mendelian randomisation (MR)
analysis, authors found Five proteins with a potential aetiological or causal role in breast cancer, providing a
basis for further functional evaluation of their potential as drug targets.

Response: We are thankful for the very positive and supportive comments. We agree that the specificity of the
protein assay provides high-confidence cis-pQTL, which in turn support interpretation of the MR results.

REVIEWERS' COMMENTS

Reviewer #2 (Remarks to the Author):

I have no further comments

Reviewer #3 (Remarks to the Author):

The authors have conducted a satisfactory revision. I have no further major concerns.